# A cross-sectional study evaluating tick-borne encephalitis vaccine uptake and timeliness among adults in Switzerland

Kyra D. Zens[1,2], Vasiliki Baroutsou[1,3], Philipp Sinniger[1], Phung Lang[1]*

1 Department of Public and Global Health, Epidemiology, Biostatistics and Prevention Institute, University of Zurich, Zurich, Switzerland, 2 Institute for Experimental Immunology, University of Zurich, Zurich, Switzerland, 3 Department of Clinical Research, Faculty of Medicine, University of Basel, Basel, Switzerland

* phung.lang@uzh.ch

## Abstract

The goal of this study was to evaluate timeliness of Tick-borne Encephalitis vaccination uptake among adults in Switzerland. In this cross-sectional survey, we collected vaccination records from randomly selected adults 18–79 throughout Switzerland. Of 4,626 participants, data from individuals receiving at least 1 TBE vaccination (n = 1875) were evaluated. We determined year and age of first vaccination and vaccine compliance, evaluating dose timeliness. Participants were considered "on time" if they received doses according to the recommended schedule ± a 15% tolerance period. 45% of participants received their first TBE vaccination between 2006 and 2009, which corresponds to a 2006 change in the official recommendation for TBE vaccination in Switzerland. 25% were first vaccinated aged 50+ (mean age 37). More than 95% of individuals receiving the first dose also received the second; ~85% of those receiving the second dose received the third. For individuals completing the primary series, 30% received 3 doses of Encepur, 58% received 3 doses of FSME-Immun, and 12% received a combination. According to "conventional" schedules, 88% and 79% of individuals received their second and third doses "on time", respectively. 20% of individuals receiving Encepur received their third dose "too early". Of individuals completing primary vaccination, 19% were overdue for a booster. Among the 31% of subjects receiving a booster, mean time to first booster was 7.1 years. We estimate that a quarter of adults in Switzerland were first vaccinated for TBE aged 50+. Approximately 80% of participants receiving at least one vaccine dose completed the primary series. We further estimate that 66% of individuals completing the TBE vaccination primary series did so with a single vaccine type and adhered to the recommended schedule.

## Introduction

Tick Borne Encephalitis (TBE) is a severe central nervous system disease caused by the TBE virus and transmitted via infected ticks. TBE is among the most frequently diagnosed viral tick-borne diseases in Europe and both incidence and geographic range continue to increase

**Data Availability Statement:** Data cannot be shared publicly due to contractual constraints between Pfizer and the University of Zurich. Therefore, we must make them available upon

request, with the stipulation that the data can be used for research purposes only. Requests for data can be made directly to the corresponding author, Phung Lang (phung.lang@uzh.ch, Hirschengraben 84, 8001 Zurich, Switzerland, +41 044 634 46 72), or to Unitectra (mail@unitectra.ch, Scheuchzerstrasse 21, 8006 Zurich, Switzerland, +41 044 634 44 01).

**Funding:** This study was supported by a grant from Pfizer WI233989. The funders had no role in the study design, data collection and analysis, decision to publish, or preparation of the manuscript.

**Competing interests:** Dr. Lang reports grants and personal fees from Pfizer during the conduct of the study. Pfizer had no role in the study design, data collection and analysis, decision to publish, or preparation of the manuscript. This does not alter our adherence to PLOS ONE policies on sharing data and materials.

[1]. While there are no curative therapies for TBE, two vaccines, Encepur and FSME-Immun, are licensed and available in Europe and recommended for individuals living, working or traveling within TBE-endemic areas [1]. Both vaccines are given as a primary series of three injections followed by boosters to maintain protective antibody titers. The European Medicines Agency (EMA)-approved "conventional" vaccination schedules include doses at day 0, 1–3 months, and 9–12 months for Encepur [2], or day 0, 1–3 months, and 5–12 months for FSME-Immun [3]. An accelerated "rapid" schedule can also be used in some circumstances with doses given on days 0, 7, and 21, followed by a fourth dose 12–18 months after the third for Encepur [2], or on days 0 and 14, followed by a third dose 5–12 months after the second for FSME-Immun [3]. Following the 3 dose primary series a first booster is recommended after 3 years and then every 5 years for individuals up to age 60 and every 3 years for those 60+ [2, 3].

As with many vaccines, the response to TBE vaccination is influenced by several factors. Age of first vaccination impacts initial immunogenicity and duration of protective responses. Among individuals aged 50+, antibody titers are reduced following both primary and booster TBE vaccination [4–7] and, in addition, the persistence of TBE-specific antibodies is significantly reduced compared to younger individuals [6, 8]. Furthermore, rates of TBE vaccine failure are increased in older, compared to younger, adults [9–12]. Adherence to priming and booster vaccination schedules also influences immunogenicity and irregular vaccination for TBE has been associated with significantly increased risk of TBE disease following exposure compared to regularly vaccinated individuals in field effectiveness studies [13, 14]. Although less clear, inconsistent use of a single vaccine type, either Encepur or FSME-Immun, during priming appears to impact neutralizing antibody responses, which could, in turn, affect immunogenicity [15, 16].

Although clear guidelines for TBE vaccination are in place, compliance by individuals/healthcare providers is not known. Furthermore, the Swiss Federal Office of Public Health (FOPH) made major changes to the official recommendation for TBE vaccination in 2006, recommending vaccination for all individuals over 6 years of age in endemic areas and extending the EMA-approved booster interval of 3–5 years, depending on age [2, 3], to 10 years for all individuals [17, 18]. How this change may have impacted vaccination uptake is unclear. Such information, however, is highly relevant for vaccination strategies and could be used to improve effectiveness. The goal of this study was to evaluate adult TBE vaccination uptake in Switzerland, potentially identifying areas for improvement.

## Methods

To determine TBE vaccination coverage we conducted a national, cross-sectional study based on obtaining vaccination records by mail [19, 20]. Adults with a Swiss mailing address in each of three age groups (18–39, 40–59, 60–79) were selected from each of the 7 Swiss geographical "large regions", defined by the Swiss Federal Statistical Office (S1 Table), by disproportional stratified random sampling. From each age group (n = 3) and region (n = 7), 1,280 individuals were invited to participate for a total sample size of 26,880 (S2 Table).

Individuals were requested twice by mail to submit a copy of their vaccination record along with a short questionnaire asking them to indicate their age and whether or not they had been vaccinated for TBE. With each mailing, a letter explaining the study's procedures and objectives was included. In this letter, individuals were informed that study participation was voluntary and that they had the possibility to withdraw submitted data at any time. They were informed that, by submitting completed questionnaires and/or vaccination records, they were consenting to participation in the study. All data were treated confidentially and anonymized

**Table 1. Study participants by age group and number of TBE vaccine doses.**

| Study Participants | n | % of Total | Swiss Pop. 18–79 (2018) |
|---|---|---|---|
| Vaccination Record Respondents | 4626 | 100 | 6570644 (100%) |
| 18–39 | 1357 | 29.3 | 2447156 (37.2%) |
| 40–59 | 1604 | 34.7 | 2486310 (37.8%) |
| 60–79 | 1665 | 36.0 | 1637178 (24.9%) |
| ≥1 TBE Dose | 1875 | 100 | - |
| 18–39 | 590 | 31.5 | - |
| 40–59 | 597 | 31.8 | - |
| 60–79 | 688 | 36.7 | - |
| ≥3 TBE Doses (Primary Series) | 1546 | 100 | - |
| 18–39 | 438 | 28.3 | - |
| 40–59 | 506 | 32.7 | - |
| 60–79 | 602 | 38.9 | - |
| ≥3 TBE Doses + Vaccine Type | 1178 | 100 | - |
| 18–39 | 346 | 29.4 | - |
| 40–59 | 392 | 33.3 | - |
| 60–79 | 440 | 37.4 | - |

prior to analysis. The study procedure and method of consent were approved by the Office of Data Protection and the Ethics Committee of the Canton of Zurich.

Of 8,192 total responses (at least a study questionnaire was returned), 4,626 individuals submitted copies of their vaccination records and were considered study participants (Table 1). Records were manually inspected and number/date(s) of TBE immunization recorded. Data were adjusted for study design and non-response. For vaccine uptake analyses, participants with 1+ TBE vaccination (n = 1875) were evaluated. To estimate timeliness, individuals completing the primary series (3+ vaccinations, n = 1546) with complete vaccine type information were included (n = 1178). Primary vaccinations were considered "on time" if they were received according to manufacturer's schedules approved in Switzerland ([2, 3], Table 2), ± a 15% "tolerance period" to provide some flexibility for slightly early or delayed doses (Table 2). Booster vaccinations were considered "on time" if they were received within the 10-year interval recommended in Switzerland [17, 18] plus a 15% "tolerance period". To convert values for months into days, we assumed 30.4 days per month (365 days/12 months). As schedules for Encepur and FSME-Immun differ, we selected the vaccine used to complete 2 or more of 3 priming doses for analysis. Analyses were performed using STATAIC16 and Prism 8. P values <0.05 were considered statistically significant.

## Results

Comparison of the year of first TBE vaccination among study participants showed a striking increase in uptake from 2006–2009, with 45% (95% CI 43–47%) of participants receiving their first dose (Fig 1A). Mean age of first vaccination was 37 (95% CI 36.7–37.6%). By age group: the mean age of first vaccination was 22 (95% CI 21.4–22.8) for those 18–39, 40 (95% CI 39.5–40.9) for those 40–59 and 58 (57.9–59.1) for those 60–79. 25% (CI 24–27%) of participants received their first vaccination aged 50+ (Fig 1B). Fig 1C shows that 96% (95% CI 95–97%) of individuals receiving the first dose also received the second, with a median time to vaccination of 34 days (95% CI 33–35, Table 3). 82% (95% CI 81–84%) of individuals receiving the second dose also received the third (median time to vaccination 287 days 95% CI 282–294, Table 3).

**Table 2. TBE vaccine dosage schedule.**

| Dosage Schedule | Lower Range[a] (days) | Upper Range[b] (days) | Tolerance Period Lower Range[c] (days) | Tolerance Period Upper Range[d] (days) |
|---|---|---|---|---|
| **"Rapid" Schedule** | | | | |
| Encepur Dose 1–2 | 7 | 7 | 6 | 8 |
| FSME-Immun Dose 1–2 | 14 | 14 | 12 | 16 |
| Encepur Dose 1–3 | 21 | 21 | 18 | 24 |
| FSME-Immun Dose 1–3 | 152 | 365 | 129 | 420 |
| **"Conventional" Schedule** | | | | |
| Encepur Dose 1–2 | 14 | 91 | 12 | 105 |
| FSME-Immun Dose 1–2 | 30 | 91 | 26 | 105 |
| Encepur Dose 2–3 | 274 | 365 | 233 | 420 |
| FSME-Immun Dose 2–3 | 152 | 365 | 129 | 420 |
| **Booster Vaccination** | | | | |
| Encepur Dose 3–4+ | 3650 | 3650 | 3103 | 4198 |
| FSME-Immun Dose 3–4+ | 3650 | 3650 | 3103 | 4198 |

[a]Lower limit, in days, of suggested time range for indicated vaccine, schedule, and dose.

[b]Upper limit, in days, of suggested time range for indicated vaccine, schedule, and dose.

[c]Lower limit, in days, of suggested time range for indicated vaccine, schedule, and dose minus an additional 15% "tolerance" period for early doses.

[d]Upper limit, in days, of suggested time range for indicated vaccine, schedule, and dose plus an additional 15% "tolerance" period for delayed doses.

We observed no difference between median or mean time to second or third doses between age groups (Table 3).

We further evaluated vaccine usage and adherence to recommended schedules. Among individuals completing the 3-dose primary series, 29% (95% CI 26–31%) received 3 doses of Encepur, and 59% (95% CI 56–62%) received 3 doses of FSME-Immun; 12% (95% CI 10–14%) received a combination. We considered that both vaccines could be administered using either a "rapid" or a "conventional" schedule [2, 3]. We found that, for both Encepur and FSME-Immun, 5% (95% CI 4–6%) of individuals followed "rapid" vaccination schedules (Fig 1D). Of the remaining individuals following "conventional" vaccination schedules, 88% (95% CI 86–91%) received the second dose "on time", and no difference was observed between vaccine types (Fig 1D). Compared to the second dose, significantly fewer individuals received the third dose "on time" (78%, 95% CI 76–81%, Fig 1D). Notably, 21% (95% CI 17–25%) of individuals receiving Encepur according to the "conventional" schedule received their third dose "too early" (sooner than 233 days after the second dose, Table 2) compared to FSME-Immun, where only 6% (95% CI 4–8%) of recipients received their third dose "too early" (sooner than 129 days after the second dose, Table 2, Fig 1D). Evaluating both vaccine types and both schedules together, we found that 66% (95% CI 63–69%) of individuals who completed the primary series did so using a single vaccine type for all priming doses and completed all priming doses "on time".

We then assessed uptake and timeliness of TBE booster vaccinations. Of individuals completing the primary series, 31% (95% CI 28–33%) received 1 or more booster(s). 19% (95% CI 17–22%) were overdue for a booster. Among those receiving a booster, the median time between completion of the primary series and the first booster was 2647 days (95% CI 2199–3137, 7.3 years, Table 3). Second boosters received were a median of 1451 days (95% CI 1167–1837, 4.0 years, Table 3) days after the first booster. We did not observe a difference between age groups in median time to first or second boosters (Table 3). We also evaluated the timing of boosters received before and after the 2006 Swiss FOPH recommendation to extend TBE

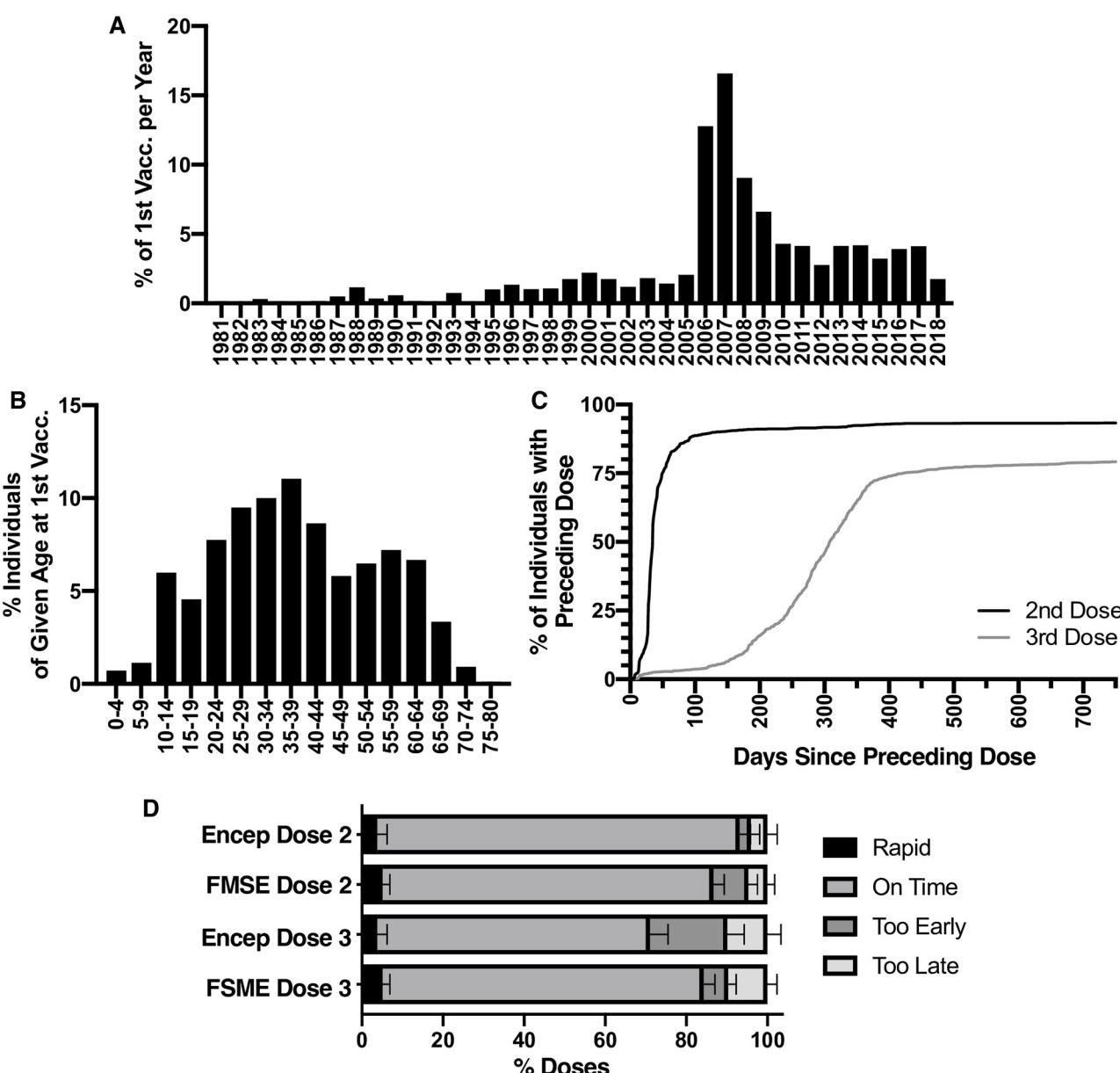

**Fig 1. Year and age of first TBE vaccination.** A) Year of First Vaccination: The frequency of individuals which received their first TBE vaccine dose in a given year, of individuals that received at least one TBE vaccine dose (n = 1861, 14 values were excluded because of missing date information in the vaccination record). Note that 2018 was a partial year (study year). B) Age at First Vaccination: The frequency of individuals which received their first TBE vaccine dose at a given age, of individuals that received at least one TBE vaccine dose (n = 1861, 14 values were excluded because of missing date information in the vaccination record, mean = 37.2, 95% CI 36.7–37.6). C) TBE Vaccine Uptake: The percentage of individuals receiving the preceding dose which also receive the subsequent dose. (n = 1854, 21 values were excluded because of missing date information for one or more doses in the vaccination record). D) Timeliness of Primary Vaccination by Vaccine Type: Among individuals completing the primary 3 dose TBE vaccination series with either majority Encepur (n = 394) or FSME-Immun (n = 745), the percentage of 2nd or 3rd doses that were received as part of the "rapid" schedule, On Time, Too Early, or Too Late, based on the manufacturer's recommendation plus or minus a 15% "tolerance period", which is intended to provide some flexibility in the analysis of slightly early or delayed doses (Table 1). Error bars represent one-sided 95% confidence intervals. Compared to the second dose, significantly fewer individuals received the third dose "on time" (78%, CI 76–81%, on time for dose 3 vs 88%, CI 86–91%, on time for dose 2, p<0.0001, Fisher's exact test). 21% (CI 76–81%) of individuals receiving Encepur received their third dose "too early" compared to 6% (CI 4–8%) of those receiving FSME-Immun (p<0.0001, Fisher's exact test).

**Table 3. Timing of TBE vaccine uptake.**

| Uptake of Vaccine Doses | n | Median days (95% CI) | Mean days (95% CI) | Range days |
|---|---|---|---|---|
| Dose 2 (of those with 1st dose) | 1769 | 35 (34–35) | 110 (86–134) | 2–5871 |
| 18–39 | 552 | 35 (33–35) | 120 (74–166) | 7–4429 |
| 40–59 | 575 | 35 (33–35) | 90 (56–124) | 2–5871 |
| 60–79 | 646 | 34 (33–35) | 123 (89–158) | 5–4569 |
| Dose 3 (of those with 2nd dose) | 1525 | 287 (282–294) | 418 (378–457) | 2–7909 |
| 18–39 | 431 | 292 (284–304) | 424 (348–501) | 11–7909 |
| 40–59 | 505 | 286 (281–301) | 417 (358–477) | 9–7311 |
| 60–79 | 593 | 280 (275–291) | 408 (348–468) | 2–5971 |
| 1st Booster (of those with 3rd dose) | 482 | 2647 (2199–3137) | 2590 (2449–2732) | 6–9868 |
| 18–39 | 118 | 2641 (1808–3264) | 2723 (2318–3004) | 9–9868 |
| 40–59 | 146 | 2722 (2041–3388) | 2550 (2329–2820) | 6–7212 |
| 60–79 | 218 | 2610 (2182–3323) | 2534 (2369–2758) | 13–7283 |
| 2nd Booster (of those with 1st booster) | 156 | 1451 (1167–1837) | 1990 (1761–2220) | 19–5931 |
| 18–39 | 36 | 1699 (1079–2636) | 2027 (1522–2531) | 27–5136 |
| 40–59 | 43 | 1183 (1144–1836) | 1917 (1505–2427) | 19–5931 |
| 60–79 | 77 | 1459 (1156–1950) | 1938 (1618–2258) | 25–4886 |

booster intervals to 10 years for all individuals. We found that the median time between completion of the primary series and the first booster prior to 2006 was 1136 days (95% CI 1119–1198, 3.1 years) and 3528 days after 2006 (95% CI 3410–3583, 9.7 years; p<0.0001 pre-2006 to post-2006, Wilcoxon Rank-Sum test). Second boosters prior to 2006 received were a mean of 1128 days (95% CI 1100–1167, 3.1 years) after the first booster and, after 2006, 2493 days (95% CI 1859–3604, 6.8 years; p<0.0001 pre-2006 to post-2006, Wilcoxon Rank-Sum test) after the first booster (S3 Table).

## Discussion

Here, we found that 25% of adults received their first vaccination aged 50+. Additionally, among adults aged 60–79, the mean age of first vaccination was 58. It is generally thought that older individuals (50+) tend not to respond as robustly to vaccination as younger individuals in terms of both 1) the overall magnitude of the immune response following vaccination, and, 2) the duration of protective immunity elicited by vaccination. It has also been shown, specifically for TBE, that age of first vaccination is associated with reduced immune responsiveness. Reduced antibody titers following both primary and booster vaccination have been observed in individuals 50+ compared to younger adults [4–7]. In addition, the persistence of TBE antibodies was significantly reduced among those aged 50+ compared to younger individuals, suggesting that the duration of protection against TBE infection following vaccination may decrease with age [6, 8]. This is supported by further work demonstrating increased rates of TBE vaccine failure in older, compared to younger, adults [9–12]. It is worth noting that, in Switzerland, TBE incidence and the risk of severe disease increase with age, with those between 60 and 75 years most affected [21], precisely when vaccination against TBE may be less effective. Interestingly, first vaccinations also increased sharply in 2006, when Switzerland officially recommended TBE vaccination for individuals 6+ in many parts of the country [17], suggesting this policy change prompted many individuals, including older adults, to receive TBE vaccination.

Vaccination compliance can be evaluated by uptake and timeliness. While most study participants receiving one TBE vaccine dose also received the second, uptake dropped between

the second and third doses. Ultimately, approximately 20% of adults in Switzerland beginning the primary series, do not complete it. Furthermore, nearly 90% and 80% of participants were "on time" for second and third doses, respectively. In Austria, where the TBE virus is highly endemic, approximately 85–90% of the population aged 16+ had received at least one dose and 50–60% had received the full primary series according to the recommended schedule [13]. In a study of self-reported TBE vaccine uptake across Central Europe, an average of 25% of respondents reported being vaccinated against TBE and 61% reported having received at least 3 doses [22]. Coverage with at least one dose was highest (outside of Austria) in Latvia (53%) and lowest in Finland and Slovakia (approximately 10%), whereas compliance with the recommended vaccination schedule was highest in Poland (97%) and lowest in Latvia and Germany (approximately 50%) [22]. In a separate Swedish study of self-reported TBE vaccine uptake, 49% of study respondents aged 18–59 and 54% of those aged 60+ reported being vaccinated against TBE, where 31% and 43% of these, respectively, reported receiving 3 or more doses [23]. In two additional German studies, just over half of individuals initiating TBE vaccination completed the three-dose schedule [24] and less than one-third were vaccinated on schedule [25]. In studies of other, non-TBE, adult vaccinees in the US and UK, compliance ranged between 30–50% [26, 27]. Whether the relatively high TBE vaccine compliance observed here extends to other vaccines in Switzerland, though, is unclear.

We further evaluated vaccine usage and adherence to recommended schedules by vaccine type. Among individuals completing the primary series, 12% received a combination of both Encepur and FSME-Immun. Importantly, sequence differences in the Envelope (E) glycoprotein [28], the major viral antigenic determinant, between the vaccine strains K23, used to manufacture Encepur, and Neudorfli, used to manufacture FSME-Immun, seem to differentially impact neutralizing antibody responses in some cases [15, 16]. How this influences immunogenicity, however, is not completely understood and how antigenic differences between the two vaccines affect interchangeability remains an area of investigation. Currently, it is generally considered acceptable to exchange vaccines in the primary series, although not desirable as efficacy has not been studied for all vaccine combinations [1, 29, 30]. In considering vaccine schedules, we found that approximately 20% of individuals receiving Encepur for their primary series received their third dose "too early" (sooner than 233 days after the second dose in our analysis). This becomes only 5%, however, if the FSME-Immun schedule is applied (the third dose of FSME-Immun can be given as soon as 129 days after the second dose in our analysis and still be "on time"). According to the approved "conventional" schedules for both vaccines, the third dose of FSME-Immun can be given as early as 5 months following the second dose, whereas the third dose of Encepur should be given no earlier than 9 months after the second [2, 3]. These results suggest that schedules for both vaccines are being used interchangeably. While it is unlikely "early" administration of Encepur negatively impacts immunogenicity, it indicates confusion regarding the vaccination schedule. When considering both "on time" administration and the use of a single vaccine type during priming, we found that 66% of participants met these combined criteria. As data support some interchangeability between vaccines [1, 29, 30], a unified recommendation for both vaccines may be warranted.

The "conventional" schedules for both Encepur and FSME-Immun suggested by the manufactures include an initial booster 3 years after completion of the primary series and every 3–5 years thereafter depending on age. In 2006, based on data demonstrating prolonged seropositivity in immunized individuals, and, in an effort to improve vaccination uptake, the Swiss FOPH put into place a uniform recommendation for TBE booster vaccination every 10 years [17, 18]. In our study, the mean times to both the first and second boosters among participants (7.1 and 5.5 years, respectively) were longer than manufacturer's recommendations, but met Swiss recommendations [18]. We did observe a significant difference, however, in the timing

of booster vaccination before and after the 2006 recommendation change to extend booster intervals with a median time between completion of the primary series and the first booster of 3.1 years prior to 2006 and 9.7 years after 2006; second boosters were received a median of 3.1 years after the first booster prior to 2006 and 6.8 years after the first booster after 2006. Although second boosters were received sooner than first boosters, these findings suggest that the national recommendation change has led to an increase in booster intervals as intended. Overall, however, approximately 1 in 5 participants completing the primary series had not received a booster in the last 10 years and was "overdue" suggesting a need for interventions to promote booster immunizations.

An important limitation in this study was the participation rate. Approximately 17% of contacted individuals submitted a vaccination record, which is low, but not unusual for a study design based on response by mail. Such a study design comes with the risk that individuals interested in a topic are more likely to participate. From an analysis of all 8,192 responses that we received (including individuals who did not submit vaccination records and were not considered study participants), we found that individuals who self-reported being vaccinated for TBE were more likely to also submit a vaccination record compared to those who reported not being vaccinated (42% versus 35% $p<0.0001$, Rao-Scott Chi-Square test), indicating some level of bias. Additionally, participation was highest among individuals aged 60–79 (36.0%) and lowest among those 18–39 (29.3%) although they represent 24.9% and 37.2% of the adult Swiss population, respectively. To control for these demographic differences, data were adjusted for study design and non-response and post-stratified. However, as the schedule for TBE vaccination in Switzerland is the same for all individuals, we do not anticipate that low response rate or differences in response by age should impact our evaluation of timeliness or vaccine uptake. In other European countries where TBE vaccination is performed according to manufacturer's schedules, which suggest that individuals 60+ be vaccinated every 3 years, compared to every 5 years for those under 60, this may differ.

In this study we demonstrate comparably high adult TBE vaccination compliance in Switzerland, though a substantial proportion of individuals were first vaccinated at an "advanced" age. A 2014 report estimated 4% vaccine failure among Swiss TBE cases [31]. We propose that studies of how irregular TBE vaccination and advanced age of first vaccination impact vaccine effectiveness are warranted to better inform vaccination policy decisions.

## Supporting information

**S1 Table. Swiss large geographic regions and integrated cantons.**
(DOCX)

**S2 Table. Determination of study sample size.**
(DOCX)

**S3 Table. Timing of TBE booster vaccine uptake prior to and after 2006.**
(DOCX)

## Acknowledgments

We would like to thank Anna Fraefel, Carlotta Superti-Furga, and Stefan Olarte for their help with data collection and organization during the study.

## Author Contributions

**Formal analysis:** Kyra D. Zens, Vasiliki Baroutsou, Philipp Sinniger.

**Funding acquisition:** Phung Lang.

**Project administration:** Phung Lang.

**Resources:** Phung Lang.

**Visualization:** Kyra D. Zens.

**Writing – original draft:** Kyra D. Zens.

**Writing – review & editing:** Kyra D. Zens, Vasiliki Baroutsou, Philipp Sinniger, Phung Lang.

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
