## [Decision Letter · Decision Letter 0]

18 Jun 2021

PONE-D-21-05494

A Cross-Sectional Study Evaluating Tick-borne Encephalitis Vaccine Uptake and Timeliness Among Adults in Switzerland

PLOS ONE

Dear Dr. Lang,

Thank you for submitting your manuscript to PLOS ONE. After careful consideration, we feel that it has merit but does not fully meet PLOS ONE’s publication criteria as it currently stands. Therefore, we invite you to submit a revised version of the manuscript that addresses the points raised below during the review process.

We look forward to receiving your revised manuscript.

Kind regards,

Ray Borrow, Ph.D., FRCPath

Academic Editor

PLOS ONE

Journal Requirements:

2. In the Methods section, please provide additional details regarding how 7 Swiss regions were selected.

Furthermore, in your Methods section, please provide a justification for the sample size used in your study, including any relevant power calculations (if applicable).

3. In the ethics statement we have note that "Submission of the vaccination records and/or

completed questionnaires were taken as informed consent." Please clarify whether IRB approved for this method of informed consent procedure, and please provide clarification as to why written informed consent could not be obtained.

5. Thank you for stating the following in the Financial Disclosure section:

'This study was supported by a grant from Pfizer WI233989. The funders had no role in the study design, data collection and analysis, decision to publish, or preparation of the manuscript.'

We note that you received funding from a commercial source: [Name of Company]

Additional Editor Comments (if provided):

Reviewers' comments:

Reviewer's Responses to Questions

**Comments to the Author**

1. Is the manuscript technically sound, and do the data support the conclusions?

Reviewer #1: Yes

Reviewer #2: Yes

2. Has the statistical analysis been performed appropriately and rigorously? 

Reviewer #1: Yes

Reviewer #2: Yes

3. Have the authors made all data underlying the findings in their manuscript fully available?

Reviewer #1: No

Reviewer #2: Yes

4. Is the manuscript presented in an intelligible fashion and written in standard English?

Reviewer #1: Yes

Reviewer #2: Yes

5. Review Comments to the Author

Reviewer #1: This is an ambitious and interesting paper with mainly local importance for the Canton of Zurich, but that that can inspire to similar local studies in other geographic areas.

All data is not availble due to participant confidentiality and can only be retrieved following approval by the office of Data Protection and the Ethic Commitee of the Canton of Zurich.

I have some concerns regarding the statistics of fig 1b, where the text states “ 96% of the individuals receiving the first dose also received the second , with a mean time to vaccination of 110 days ( 95% CI87-134, median 34 days 95%CI33-35)”. There is a big discrepancy between mean and median time including CI. If the numbers are correct I would recommend to use only median, using both is mostly confusing.

Apart from this I would appreciate an extended discussion including:

a. Is there a risk for bias when only 17% of individuals submitted their vaccine records?

b. What is the importance of different mean age of first vaccination in different age groups?

c. You state that 66% of participants followed the “ideal schedule”. Is the “ideal schedule” of importance? What research is there to support this or another schedule?

d. You report that the second booster dose was given earlier than the first booster dose. I am curious why this might be? Do you have a suggestion/theory?

Reviewer #2: "Tolerance interval" - in Abstract, Methods section, Figure 1 as well - please describe and provide references for this term, how it applies to vaccination schedules, analysis.

Introduction:

"Age of first vaccination impacts initial immunogenicity and duration of protective responses" - not sure this is true and hard evidence based. Please provide references and how it applies to all vaccines, espec. TBE

Sentence "TBE has two vaccine formulations..."

1) two vaccine formulations in EU registered, but overall there are more

2) WHO Position paper states that both (EU) are interchangeable

"The goal of this study... uptake in Switzerland..." - reference for vaccination against TBE guidelines in Switzerland.

Methods:

Please make clear the schedules which are the recommended in the country (SmPC?) and used for study purposes

Should be made clear, that 10 y interval in Switzerland is not in compliance with vaccine registration details

Table 2 is difficult to understand

Results:

- mean time between dose 1 and dose 2 - is VERY long. Please reflect on this.

- please describe what has been meant by "rapid" and "conventional" (by SmPC?)

- "...Encepur received their third dose "too early"..." - what does it mean "too early"

- "ideal" schedule - there are no such ideal and nonideal schedules. There are registered, approved schedules, but not "ideal" and non ideal"

Discussion:

"Age of first TBE vaccination impacts immune responsiveness and vaccine failures...." - Why? Not just because of the first vaccination at certain age, also commodities in those 50+.

Please make clear what authors really mean - vaccine failure or ineffectiveness. Those are different terms and describe different reasons for certain outcomes.

In the paragraph dedicated to vaccination compliance - comparison with compliance in TBE endemic countries

Interchangeability - main reference should be WHO Position paper, not a 2006 single reference.

WHO Position Paper is not at all cited.

6. PLOS authors have the option to publish the peer review history of their article (what does this mean?). If published, this will include your full peer review and any attached files.

Reviewer #1: No

Reviewer #2: **Yes: **Dace Zavadska

---

## [Author Response · Author response to Decision Letter 0]

29 Oct 2021

Journal Requirements:

We have modified our manuscript to meet the style requirements outlined.

2. In the Methods section, please provide additional details regarding how 7 Swiss regions were selected.

The 7 Swiss large regions are those subdivisions defined by the Nomenclature of Territorial Units for Statistics (NUTS) developed by the European Union.

https://ec.europa.eu/eurostat/web/nuts/nuts-maps

https://ec.europa.eu/eurostat/documents/345175/7451602/2021-NUTS-2-map-CH.pdf

We selected these regions in order to achieve coverage of the entire country, but with a manageable sample size.

We have modified the text as follows and included a supporting table (S1 Table) to clarify that these are defined regions and not generated for the purpose of this study: “Adults with a Swiss mailing address in each of three age groups (18-39, 40-59, 60-79) were selected from each of the 7 Swiss geographical ‘large regions’, defined by the Swiss Federal Statistical Office (S1 Table),” (page 4, lines 81-83)

Furthermore, in your Methods section, please provide a justification for the sample size used in your study, including any relevant power calculations (if applicable).

We have added a supporting table (S2 Table) where we provide our sample size calculation and assumptions made. We have modified the text as follows: “From each age group (n=3) and region (n=7), 1,280 individuals were invited to participate for a total sample size of 26,880 (S2 Table).” (page 5, lines 84-85)

3. In the ethics statement we have note that "Submission of the vaccination records and/or

completed questionnaires were taken as informed consent." Please clarify whether IRB approved for this method of informed consent procedure, and please provide clarification as to why written informed consent could not be obtained.

Yes, the entire study procedure, including that submission of vaccination records was taken as informed consent, was approved by the Department of Data Protection at the University of Zurich as well as the Ethics Committee of the Canton of Zurich. We agree, though, that this statement was ambiguous and have added the following details/language to better clarify: “Individuals were requested twice by mail to submit a copy of their vaccination record along with a short questionnaire asking them to indicate their age and whether or not they had been vaccinated for TBE. With each mailing, a letter explaining the study's procedures and objectives was included. In this letter, individuals were informed that study participation was voluntary and that they had the possibility to withdraw submitted data at any time. They were informed that, by submitting completed questionnaires and/or vaccination records, they were consenting to participation in the study. All data were treated confidentially and anonymized prior to analysis. The study procedure and method of consent were approved by the Office of Data Protection and the Ethics Committee of the Canton of Zurich.” (page 5, lines 87-95)

4. We note that you have indicated that data from this study are available upon request. PLOS only allows data to be available upon request if there are legal or ethical restrictions on sharing data publicly.

The contract between Pfizer, which funded the submitted study, and the University of Zurich includes the following language:

“Study Data and Study Results. For purposes of this Agreement, ‘Study Data’ means the raw, non-aggregated data collected about each Study subject during the course of the Study. ‘Study Results’ refers to aggregated or summarized Study Data and conclusions about the Study, as would be included in a study report or publication. Principal Investigator is free to publish the Study Results, subject to the provisions in Section 7 (Publications) and Principal Investigator and Institution are free to use Study Results for any other purpose. Institution owns and is free to use the Study Data for its own research and educational, and patient care purposes and programs. However, in consideration of the Pfizer Investigator Initiated Research Support, Principal Investigator and Institution will not use or permit others to use the Study Data for the commercial benefit of any third party.”

Both Pfizer and the University of Zurich have agreed that, as the raw data would be publicly available, they could be used for commercial benefit of a third party. Therefore, we must make them available upon request with the stipulation that the data can be used for research purposes only. Requests for data can be made directly to the corresponding author, Phung Lang (phung.lang@uzh.ch, Hirschengraben 84, 8001 Zurich, Switzerland, +41 044 634 46 72), or to Unitectra (mail@unitectra.ch, Scheuchzerstrasse 21, 8006 Zurich, Switzerland, +41 044 634 44 01).

5. Thank you for stating the following in the Financial Disclosure section: 'This study was supported by a grant from Pfizer WI233989. The funders had no role in the study design, data collection and analysis, decision to publish, or preparation of the manuscript.' We note that you received funding from a commercial source. Please provide an amended Competing Interests Statement that explicitly states this commercial funder, along with any other relevant declarations relating to employment, consultancy, patents, products in development, marketed products, etc.

Please find our amended Competing Interests Statement as follows:

Dr. Lang reports grants and personal fees from Pfizer during the conduct of the study. Pfizer had no role in the study design, data collection and analysis, decision to publish, or preparation of the manuscript. This does not alter our adherence to PLOS ONE policies on sharing data and materials. 

Reviewer Comments to the Author

Reviewer #1: This is an ambitious and interesting paper with mainly local importance for the Canton of Zurich, but that that can inspire to similar local studies in other geographic areas.

All data is not available due to participant confidentiality and can only be retrieved following approval by the office of Data Protection and the Ethic Committee of the Canton of Zurich.

Unfortunately, as the contract funding this study specifies that the “…Principal Investigator and Institution will not use or permit others to use the Study Data for the commercial benefit of any third party.”, we are unable to make our data publicly available as we cannot guarantee that they will not be used for the commercial benefit of a third party. This has been confirmed by both Pfizer and the University of Zurich, and, therefore, we must make them available upon request with the stipulation that the data can be used for research purposes only.

I have some concerns regarding the statistics of fig 1b, where the text states “ 96% of the individuals receiving the first dose also received the second , with a mean time to vaccination of 110 days ( 95% CI87-134, median 34 days 95%CI33-35)”. There is a big discrepancy between mean and median time including CI. If the numbers are correct I would recommend to use only median, using both is mostly confusing.

Thank you for this helpful feedback. We agree that in this case it makes the most sense to present the median values. We have modified the text to include only the median dates, but have included the mean dates with 95% confidence intervals as well as the range of dates in a revised version of Table 3. The discrepancy between mean and median values is due to a small number of individuals with prolonged periods between doses (more than 1-2 decades, in some cases). 

The revised text is as follows: “Comparison of the year of first TBE vaccination among study participants showed a striking increase in uptake from 2006-2009, with 45% (95% CI 43-47%) of participants receiving their first dose (Fig 1a). Mean age of first vaccination was 37 (95% CI 36.7-37.6%). By age group; the mean age of first vaccination was 22 (95% CI 21.4-22.8) for those 18-39, 40 (95% CI 39.5-40.9) for those 40-59 and 58 (57.9-59.1) for those 60-79. 25% (CI 24-27%) of participants received their first vaccination aged 50+ (Fig 1b). Fig 1c shows that 96% (95% CI 95-97%) of individuals receiving the first dose also received the second, with a median time to vaccination of 34 days (95% CI 33-35, Table 3). 82% (95% CI 81-84%) of individuals receiving the second dose also received the third (median time to vaccination 287 days 95% CI 282-294, Table 3). We observed no difference between median or mean time to second or third doses between age groups (Table 3).” (page 7, lines 126-136)

“We then assessed uptake and timeliness of TBE booster vaccinations. Of individuals completing the primary series, 31% (95% CI 28-33%) received 1 or more booster(s). 19% (95% CI 17-22%) were overdue for a booster. Among those receiving a booster, the median time between completion of the primary series and the first booster was 2647 days (95% CI 2199-3137, 7.3 years, Table 3). Second boosters were received a median of 1451 days (95% CI 1167-1837, 4.0 years, Table 3) days after the first booster. We did not observe a difference between age groups in median time to first or second boosters (Table 3).” (page 10, lines 179-185)

Apart from this I would appreciate an extended discussion including:

a. Is there a risk for bias when only 17% of individuals submitted their vaccine records?

Thank you for this point – we have added the following paragraph considering study limitations to the discussion: “An important limitation in this study was the participation rate. Approximately 17% of contacted individuals submitted a vaccination record, which is low, but not unusual for a study design based on response by mail. Such a study design comes with the risk that individuals interested in a topic are more likely to participate. From an analysis of all 8,192 responses that we received (including individuals who did not submit vaccination records and were not considered study participants), we found that individuals who self-reported being vaccinated for TBE were more likely to also submit a vaccination record compared to those who reported not being vaccinated (42% versus 35% p<0.0001, Rao-Scott Chi-Squared test), indicating some level of bias. Additionally, participation was highest among individuals aged 60-79 (36.0%) and lowest among those 18-39 (29.3%) although they represent 24.9% and 37.2% the adult Swiss population, respectively. To control for these demographic differences, data were adjusted for study design and non-response. However, as the schedule for TBE vaccination in Switzerland is the same for all individuals, we do not anticipate that low response rate or differences in response by age should impact our evaluation of timeliness or vaccine uptake. In other European countries where TBE vaccination is performed according to manufacturer’s schedules, which suggest that individuals 60+ be vaccinated every 3 years, compared to every 5 years for those under 60, this may differ.” (pages 14-15, lines 274-290)

b. What is the importance of different mean age of first vaccination in different age groups?

This is an important point - Older individuals tend not to response as robustly to vaccination as younger individuals, both in terms of 1) the magnitude of the response to vaccination, and, 2) the long-term duration of protective immunity elicited by vaccination. Several serological studies have shown that this also appears to be the case for TBE vaccination. 

We present the mean age of first vaccination by age group here as a way to highlight that those 60+, who are already most at-risk for TBE infection (likely due to a combination of health factors as well as increased risk of exposure), also tended to be first vaccinated at a much older age, which could potentially impact the effectiveness of vaccination, putting them at additional risk. As Switzerland has a unique, 10-year booster schedule (without a first, 3 year booster) independent of age, we feel that it is important to point out this difference in demographics.

However, we feel that a direct comparison of ages between age groups is likely confusing and would highlight more the difference in when the vaccine became available and wide-spread vaccination was initiated in the population. To this point, we have modified the following text in the results:

“Mean age of first vaccination was 37 (95%CI 36.7-37.6%). By age group; the mean age of first vaccination was 22 (95%CI 21.4-22.8) for those 18-39, 40 (95%CI 39.5-40.9) for those 40-59 and 58 (57.9-59.1) for those 60-79. 25% (CI 24-27%) of participants received their first vaccination aged 50+ (Fig 1b).” (page 7, lines 128-131)

We have also added the following text to the discussion: “Here, we found that 25% of adults received their first vaccination aged 50+. Additionally, among adults aged 60-79, the mean age of first vaccination was 58. It is generally thought that older individuals (50+) tend not to respond as robustly to vaccination as younger individuals in terms of both 1) the overall magnitude of the immune response following vaccination, and, 2) the duration of protective immunity elicited by vaccination. It has also been shown, specifically for TBE, that age of first vaccination is associated with reduced immune responsiveness. Reduced antibody titers following both primary and booster vaccination have been observed in individuals 50+ compared to younger adults [4-7]. In addition, the persistence of TBE antibodies was significantly reduced among those aged 50+ compared to younger individuals, suggesting that the duration of protection against TBE infection following vaccination may decrease with age [6, 8]. This is supported by further work demonstrating increased rates of TBE vaccine failure in older, compared to younger, adults [9-12]. It is worth noting that, in Switzerland, TBE incidence and the risk of severe disease increase with age, with those between 60 and 75 years most affected [21], precisely when vaccination against TBE may be less effective.” (page 11, lines 195-208)

c. You state that 66% of participants followed the “ideal schedule”. Is the “ideal schedule” of importance? What research is there to support this or another schedule?

Our intention was to point out that only 66% of individuals are vaccinated on time and comply with the use of a single vaccine during priming. Our use of the word “ideal” here, then, is somewhat arbitrary, and we have revised this in the text for clarity.

In the abstract the revised text is a follows: “We further estimate that 66% of individuals completing the TBE vaccination primary series did so with a single vaccine type and adhered to the recommended schedule.” (page 2, lines 38-40)

We have modified the results as follows: “Evaluating both vaccine types and both schedules together, we found that 66% (95%CI 63-69%) of individuals who completed the primary series did so using a single vaccine type for all priming doses and completed all priming doses (including the rapid schedule) “on time”.” (page 10, lines 175-177)

We have also including the following text into the discussion: “We further evaluated vaccine usage and adherence to recommended schedules by vaccine type. Among individuals completing the primary series, 12% received a combination of both Encepur and FSME Immun. Importantly, sequence differences in the Envelope (E) glycoprotein [28], the major viral antigenic determinant, between the vaccine strains K23, used to manufacture Encepur, and Neudorfli, used to manufacture FSME-Immun, seem to differentially impact neutralizing antibody responses in some cases [15, 16]. How this influences immunogenicity, however, is not completely understood and how antigenic differences between the two vaccines affect interchangeability remains an area of investigation. Currently, it is generally considered acceptable to exchange vaccines in the primary series, although not desirable as efficacy has not been studied for all vaccine combinations [1, 29, 30]. We considering vaccine schedules, we found that approximately 20% of individuals receiving Encepur for their primary series received their third dose “too early”. This becomes only 5%, however, if the FSME-Immun schedule is applied (5-12 months rather than 9-12 months), suggesting that schedules for both vaccines are being used interchangeably. While it is unlikely “early” administration of Encepur negatively impacts immunogenicity, it indicates confusion regarding the vaccination schedule. When considering both “on time” administration and the use of a single vaccine type during priming, we found that 66% of participants met these combined criteria. As data support some interchangeability between vaccines [1, 29, 30], a unified recommendation for both vaccines may be warranted.” (pages 12-13, lines 233-255)

d. You report that the second booster dose was given earlier than the first booster dose. I am curious why this might be? Do you have a suggestion/theory?

We believe that this is due to differences in when doses were administered in relation to the 2006 change in TBE vaccination recommendations made by the Swiss Federal Office of Public Health. Prior to 2006, the recommendation for boosters in Switzerland was as according to the manufacturers’ recommendations (3-year initial booster, followed by subsequent boosters every 5 years for those under 60 and 3 years for those 60+). In 2006 this was modified to a general recommendation for boosters every 10 years after the completion of the primary vaccination series.

In an additional analysis of our dataset, we find that there is no significant difference in the time between completion of the primary series and the first booster compared to time between the first and second booster when they were received prior to 2006. In assessing timing between primary-to-first booster and first-to-second booster after 2006, we see a significant, though less striking, difference in the timing between doses. Based on these data, we believe that, as more time elapses after the recommendation change, the period until the second or third boosters will extend (for example, if someone was vaccinated first in 2006, they would have been eligible for their first booster in 2016, but would not yet be eligible for their second booster, which would be 2026).

We have included this additional information in the results as follows: “We also evaluated the timing of boosters received before and after the 2006 Swiss FOPH recommendation to extend TBE booster intervals to 10 years for all individuals. We found that the median time between completion of the primary series and the first booster prior to 2006 was 1136 days (95% CI 1119-1198, 3.1 years) and 3528 days after 2006 (95% CI 3410-3583, 9.7 years; p<0.0001 pre-2006 to post-2006, Wilcoxon Rank-Sum test). Second boosters prior to 2006 were received a mean of 1128 days (95% CI 1100-1167, 3.1 years) after the first booster and, after 2006, 2493 days (95% CI 1859-3604, 6.8 years; p<0.0001 pre-2006 to post-2006, Wilcoxon Rank-Sum test) after the first booster (S3 Table).” (pages 10-11, lines 185-192)

The following is also included in the discussion: “The “conventional” schedules for both Encepur and FSME-Immun suggested by the manufactures include an initial booster 3 years after completion of the primary series and every 3-5 years thereafter depending on age. In 2006, based on data demonstrating prolonged seropositivity in immunized individuals, and, in an effort to improve vaccination uptake, the Swiss FOPH put into place a uniform recommendation for TBE booster vaccination every 10 years [17, 18]. In our study, the mean times to both the first and second boosters among participants (7.1 and 5.5 years, respectively) were longer than manufacturer’s recommendations, but met Swiss recommendations [18]. We did observe a significant difference, however, in the timing of booster vaccination before and after the 2006 recommendation change to extend booster intervals with a median time between completion of the primary series and the first booster of 3.1 years prior to 2006 and 9.7 years after 2006 whereas second boosters were received a median of 3.1 years after the first booster prior to 2006 and 6.8 years after the first booster after 2006. Although second boosters were received sooner than first boosters, these findings suggest that the national recommendation change has led to an increase in booster intervals as intended. Overall, however, approximately 1 in 5 participants completing the primary series had not received a booster in the last 10 years and was “overdue” suggesting a need for interventions to promote booster immunizations.” (pages 13-14, lines 257-272)

We have also included an additional supporting table (S3 Table) with these values.

Reviewer #2: "Tolerance interval" - in Abstract, Methods section, Figure 1 as well - please describe and provide references for this term, how it applies to vaccination schedules, analysis.

Here we intend the term “tolerance period” to refer to additional time outside of the recommended time range for vaccination where we still consider a vaccination to be “on time”. The concept is analogous to a “grace period”. The Centers for Disease Control and Prevention’s Pink Book (Appendix A Recommended and Minimum Ages and Intervals Between Doses of Routinely Administered Recommended Vaccines; https://www.cdc.gov/vaccines/pubs/pinkbook/downloads/appendices/a/age-interval-table.pdf) includes a grace period of 4 days prior to the recommended vaccination interval as acceptable in the interpretation of whether a vaccine was received “on time”. This concept was also applied in a study evaluating TBE vaccination compliance in Germany in 2017 (Jabob https://doi.org/10.1016/j.cmi.2017.01.012). However, the authors used a 25% tolerance period in this case.

We introduced the concept of a “tolerance period” into our analysis as we observed from our dataset that individuals were frequently vaccinated a few days on either side of the recommended range. For example, at day 28 or day 32 rather than at “1 month” (or 30 days). Furthermore, the concept of “1 month” in days, can, in practice, have different interpretations – such as 4 weeks (or 28 days), 30 days, or 31 days. We also wanted to include this flexibility as scheduling physician’s visits on exact days is not always realistic. Our intention was to include as many individuals as possible considering such reasonable limitations. Importantly, this variation – several days on either side of the recommended schedule for vaccination - likely does not negatively impact the immune response to vaccination where antibody responses are typically detectable by 2 weeks following the second immunization and are readily detectable again within 3 weeks of the third or booster immunizations (Harabacz https://doi.org/10.1016/0264-410X(92)90003-3, Zent https://doi.org/10.1016/S0264-410X(02)00592-3, Zent https://doi.org/10.1016/j.vaccine.2003.08.005, Loew-Baselli https://doi.org/10.1016/j.vaccine.2006.03.061).

We elected to use a 15% tolerance period in our analysis, rather than a “grace period” of a given number of days (such as 4 used by the CDC) as it was flexible with regard to the length of the interval between doses (for longer intervals it seems reasonable that vaccination on precise dates is less important). Furthermore, we felt that a 25% tolerance period may provide too much flexibility, especially for shorter dosing intervals (for example, a 14-day interval could be as short as 10 days considering a 25% tolerance period). We have included some brief language in the methods and results to explain this concept.

Introduction:

"Age of first vaccination impacts initial immunogenicity and duration of protective responses" - not sure this is true and hard evidence based. Please provide references and how it applies to all vaccines, espec. TBE. 

This is an important point which we have not clarified well here. That aging/immunosenescence impact immune responses to vaccination is well-accepted; for an excellent review please see Crooke et al. 2019 (doi: 10.1186/s12979-019-0164-9). This is also fairly well-studied in the context of TBE vaccination. Both the magnitude of the antibody response following vaccination, as well as the persistence of these responses are reduced in older individuals (studies have demonstrated from 50+) compared to younger individuals. Rates of vaccine failure are also higher among older individuals. We have revised this paragraph to include this very relevant information, as well as appropriate citations. The new text is as follows: “As with many vaccines, the response to TBE vaccination is influenced by several factors. Age of first vaccination impacts initial immunogenicity and duration of protective responses. Among individuals aged 50+, antibody titers are reduced following both primary and booster TBE vaccination [4-7] and, in addition, the persistence of TBE-specific antibodies is significantly reduced compared to younger individuals [6, 8]. Furthermore, rates of TBE vaccine failure are increased in older, compared to younger, adults [9-12]. Adherence to priming and booster vaccination schedules also influences immunogenicity and irregular vaccination for TBE has been associated with significantly increased risk of TBE disease following exposure compared to regularly vaccinated individuals in field effectiveness studies [13, 14].” (pages 3-4, lines 57-65)

Sentence "TBE has two vaccine formulations..."

1) two vaccine formulations in EU registered, but overall there are more

2) WHO Position paper states that both (EU) are interchangeable

These are also both excellent points which require additional clarification. We have included that two TBE vaccines are licensed and available in Europe. Although it is generally accepted that Encepur and FSME-Immun are interchangeable, it is known that there are antigenic differences between the K23 and Neudorfli strains used to manufacture the two vaccines, respectively. These differences further occur in the Envelope glycoprotein, which is the major target of the neutralizing antibody response to the TBE virus. Studies have also shown that the two vaccines induce differing neutralizing antibody responses. While evidence supports interchangeability of these two vaccines, the impact of these differences on protection from disease, also in the context of a mixed priming schedule, are not completely clear and, we feel, warrant further study. Consistent with this, we have added the following text and citations in the introduction to better explain: “Although less clear, inconsistent use of a single vaccine type, either Encepur or FSME-Immun, during priming appears to impact neutralizing antibody responses, which could, in turn, affect immunogenicity [15, 16].” (page 4, lines 65-67)

We have also included the following text in the discussion section: “Among individuals completing the primary series, 12% received a combination of both Encepur and FSME Immun. Importantly, sequence differences in the Envelope (E) glycoprotein [28], the major viral antigenic determinant, between the vaccine strains K23, used to manufacture Encepur, and Neudorfli, used to manufacture FSME-Immun, seem to differentially impact neutralizing antibody responses in some cases [15, 16]. How this influences immunogenicity, however, is not completely understood and how antigenic differences between the two vaccines affect interchangeability remains an area of investigation. Currently, it is generally considered acceptable to exchange vaccines in the primary series, although not desirable as efficacy has not been studied for all vaccine combinations [1, 29, 30].” (pages 12-13, lines 234-242)

"The goal of this study... uptake in Switzerland..." - reference for vaccination against TBE guidelines in Switzerland.

We have modified this section of the introduction as follows and have included a reference to the official recommendation change made in 2006 which forms the basis for the current TBE vaccination guidelines: “Although clear guidelines for TBE vaccination are in place, compliance by individuals/healthcare providers is not known. Furthermore, the Swiss Federal Office of Public Health (FOPH) made major changes to the official recommendation for TBE vaccination in 2006, recommending vaccination for all individuals over 6 years of age in endemic areas and extending the manufacturer-recommended booster interval of 3-5 years, depending on age, to 10 years for all individuals [17, 18]. How this change may have impacted vaccination uptake is unclear. Such information, however, is highly relevant for vaccination strategies and could be used to improve effectiveness. The goal of this study was to evaluate adult TBE vaccination uptake in Switzerland, potentially identifying areas for improvement.” (page 4, lines 69-77)

Methods: Please make clear the schedules which are the recommended in the country (SmPC?) and used for study purposes

We have included the following text in the introduction: “Both vaccines are given as a primary series of three injections followed by boosters to maintain protective antibody titers. The European Medicines Agency (EMA)-approved “conventional” vaccination schedules include doses at day 0, 1–3 months, and 9–12 months for Encepur [2], or day 0, 1–3 months, and 5–12 months for FSME-Immun [3]. An accelerated “rapid” schedule can also be used in some circumstances with doses given on days 0, 7, and 21, followed by a fourth dose 12–18 months after the third for Encepur [2], or on days 0 and 14, followed by a third dose 5–12 months after the second for FSME-Immun [3]. Following the 3 dose primary series a first booster is recommended after 3 years and then every 5 years for individuals up to age 60 and every 3 years for those 60+ [2, 3].” (page 3, lines 47-55)

The schedules used for defining timeliness are given in Table 2. The lower and upper bounds (shown in the second and third columns) give the range (in days) when each dose is indicated. Taking the first row for example, according to the rapid schedule for Encepur, the number of days between dose 1 and dose 2 should be a minimum of 7 and a maximum of 7 (there is no range for this dose according to the rapid schedule). Taking the fourth row for example, according to the conventional schedule for FSME Immun, the number of days between dose 1 and dose 2 should be a minimum of 30 and a maximum of 91. We hope that this clarifies the schedules for both vaccines and which dates were used for study purposes.

Should be made clear, that 10 y interval in Switzerland is not in compliance with vaccine registration details

Thank you for highlighting this important point – we have included text to this affect in the introduction (“Furthermore, the Swiss Federal Office of Public Health (FOPH) made major changes to the official recommendation for TBE vaccination in 2006, recommending vaccination for all individuals over 6 years of age in endemic areas and extending the EMA-approved booster interval of 3-5 years, depending on age [2, 3], to 10 years for all individuals [17, 18].”) (page 4, lines 70-74), as well as in the discussion (“In 2006, based on data demonstrating prolonged seropositivity in immunized individuals, and, in an effort to improve vaccination uptake, the Swiss FOPH put into place a uniform recommendation for TBE booster vaccination every 10 years [17, 18].”). (page 14, lines 259-261)

Table 2 is difficult to understand

We have added footnotes to the table to better indicate the values included in each row/column and modified the column and row titles to improve clarity.

Results:

- mean time between dose 1 and dose 2 - is VERY long. Please reflect on this.

The discrepancy between mean and median values is due to a small number of individuals with prolonged periods between doses (more than 1-2 decades, in some cases). Based on feedback from reviewer 1, we have modified the text to include only the median dates, but have included the mean dates with 95% confidence intervals as well as the range of dates in Table 3 (as well as the new S3 Table). 

- please describe what has been meant by "rapid" and "conventional" (by SmPC?)

We have included the following text in the introduction to clarify the distinction between “rapid” and “conventional” schedules: “The European Medicines Agency (EMA)-approved “conventional” vaccination schedules include doses at day 0, 1–3 months, and 9–12 months for Encepur [2], or day 0, 1–3 months, and 5–12 months for FSME-Immun [3]. An accelerated “rapid” schedule can also be used in some circumstances with doses given on days 0, 7, and 21, followed by a fourth dose 12–18 months after the third for Encepur [2], or on days 0 and 14, followed by a third dose 5–12 months after the second for FSME-Immun [3]. Following the 3 dose primary series a first booster is recommended after 3 years and then every 5 years for individuals up to age 60 and every 3 years for those 60+ [2, 3].” (page 3, lines 48-55)

- "...Encepur received their third dose "too early"..." - what does it mean "too early"

For the purposes of our analysis, it means sooner than 233 days after the second dose. We have added the following text to the results and discussion to clarify:

“Compared to the second dose, significantly fewer individuals received the third dose “on time” (78%, 95% CI 76-81%, Fig 1d). Notably, 21% (95% CI 17-25%) of individuals receiving Encepur according to the “conventional” schedule received their third dose “too early” (sooner than 233 days after the second dose, Table 2) compared to FSME-Immun, where only 6% (95% CI 4-8%) of recipients received their third dose “too early” (sooner than 129 days after the second dose, Table 2, Fig 1d).” (page 10, lines 170-175)

“In considering vaccine schedules, we found that approximately 20% of individuals receiving Encepur for their primary series received their third dose “too early” (sooner than 233 days after the second dose in our analysis). This becomes only 5%, however, if the FSME-Immun schedule is applied (the third dose of FSME-Immun can be given as soon as 129 days after the second dose in our analysis and still be “on time”. According to the approved “conventional” schedules for both vaccines, the third dose of FSME-Immun can be given as early as 5 months following the second dose, whereas the third dose of Encepur should be given no earlier than 9 months after the second [2, 3]. These results suggest that schedules for both vaccines are being used interchangeably. While it is unlikely “early” administration of Encepur negatively impacts immunogenicity, it indicates confusion regarding the vaccination schedule.” (page 13, lines 242-255)

- "ideal" schedule - there are no such ideal and nonideal schedules. There are registered, approved schedules, but not "ideal" and non ideal"

We have removed this language from the text and attempted to clarify this in a more explicit way. Please also see the following text copied from above (our response to point c from reviewer 1):

“Our intention was to point out that only 66% of individuals are vaccinated on time and comply with the use of a single vaccine during priming. Our use of the word “ideal” here, then, is somewhat arbitrary, and we have revised this in the text for clarity.

In the abstract the revised text is a follows: “We further estimate that 66% of individuals completing the TBE vaccination primary series did so with a single vaccine type and adhered to the recommended schedule.” (page 2, lines 38-40)

We have modified the results as follows: “Evaluating both vaccine types and both schedules together, we found that 66% (95%CI 63-69%) of individuals who completed the primary series did so using a single vaccine type for all priming doses and completed all priming doses (including the rapid schedule) “on time”.” (page 10, lines 175-177)

We have also including the following text into the discussion: “We further evaluated vaccine usage and adherence to recommended schedules by vaccine type. Among individuals completing the primary series, 12% received a combination of both Encepur and FSME Immun. Importantly, sequence differences in the Envelope (E) glycoprotein [28], the major viral antigenic determinant, between the vaccine strains K23, used to manufacture Encepur, and Neudorfli, used to manufacture FSME-Immun, seem to differentially impact neutralizing antibody responses in some cases [15, 16]. How this influences immunogenicity, however, is not completely understood and how antigenic differences between the two vaccines affect interchangeability remains an area of investigation. Currently, it is generally considered acceptable to exchange vaccines in the primary series, although not desirable as efficacy has not been studied for all vaccine combinations [1, 29, 30]. We considering vaccine schedules, we found that approximately 20% of individuals receiving Encepur for their primary series received their third dose “too early”. This becomes only 5%, however, if the FSME-Immun schedule is applied (5-12 months rather than 9-12 months), suggesting that schedules for both vaccines are being used interchangeably. While it is unlikely “early” administration of Encepur negatively impacts immunogenicity, it indicates confusion regarding the vaccination schedule. When considering both “on time” administration and the use of a single vaccine type during priming, we found that 66% of participants met these combined criteria. As data support some interchangeability between vaccines [1, 29, 30], a unified recommendation for both vaccines may be warranted.” (pages 12-13, lines 233-255)”

Discussion: "Age of first TBE vaccination impacts immune responsiveness and vaccine failures...." - Why? Not just because of the first vaccination at certain age, also commodities in those 50+. Please make clear what authors really mean - vaccine failure or ineffectiveness. Those are different terms and describe different reasons for certain outcomes.

As described, it is well-accepted that age influences the immune response to vaccination. While comorbidities may additionally play a role in this, immunosenescence is also a known, independent driver of immune dysfunction with age. There is a significant body of literature which has demonstrated that individuals 50+ mount virus-specific antibody responses of reduced magnitude following TBE vaccination, specifically. Furthermore, the duration of persistence of these responses has been shown to be reduced. As the contributions of immunosenescence versus comorbidities were not investigated in these studies, we cannot be more specific than the use of the term “age”, which is the criterion used by the authors. However, we have changed the language of our statement from “age…impacts” to “age…is associated with” to remove the implication of a causal relationship. The new discussion text is as follows:

“It is generally thought that older individuals (50+) tend not to respond as robustly to vaccination as younger individuals in terms of both 1) the overall magnitude of the immune response following vaccination, and, 2) the duration of protective immunity elicited by vaccination. It has also been shown, specifically for TBE, that age of first vaccination is associated with reduced immune responsiveness. Reduced antibody titers following both primary and booster vaccination have been observed in individuals 50+ compared to younger adults [4-7]. In addition, the persistence of TBE antibodies was significantly reduced among those aged 50+ compared to younger individuals, suggesting that the duration of protection against TBE infection following vaccination may decrease with age [6, 8]. This is supported by further work demonstrating increased rates of TBE vaccine failure in older, compared to younger, adults [9-12]. It is worth noting that, in Switzerland, TBE incidence and the risk of severe disease increase with age, with those between 60 and 75 years most affected [21], precisely when vaccination against TBE may be less effective.” (page 11, lines 196-208)

In the paragraph dedicated to vaccination compliance - comparison with compliance in TBE endemic countries

We have modified the text as follows to include information about TBE vaccination compliance in other endemic countries: “Vaccination compliance can be evaluated by uptake and timeliness. While most study participants receiving one TBE vaccine dose also received the second, uptake dropped between the second and third doses. Ultimately, approximately 20% of adults in Switzerland beginning the primary series do not complete it. Furthermore, nearly 90% and 80% of participants were “on time” for second and third doses, respectively. In Austria, where the TBE virus is highly endemic, approximately 85-90% of the population aged 16+ had received at least one dose and 50-60% had received the full primary series according to the recommended schedule [13]. In a study of self-reported TBE vaccine uptake across Central Europe, an average of 25% of respondents reported being vaccinated against TBE and 61% reported having received at least 3 doses [22]. Coverage with at least one dose was highest (outside of Austria) in Latvia (53%) and lowest in Finland and Slovakia (approximately 10%), whereas compliance with the recommended vaccination schedule was highest in Poland (97%) and lowest in Latvia and Germany (approximately 50%) [22]. In a separate Swedish study of self-reported TBE vaccine uptake, 49% of study respondents aged 18-59 and 54% of those aged 60+ reported being vaccinated against TBE and 31% and 43% of these, respectively, reported receiving 3 or more doses [23]. In two additional German studies, just over half of individuals initiating TBE vaccination completed the three-dose schedule [24] and less than one-third were vaccinated on schedule [25]. In studies of other, non-TBE, adult vaccinees in the US and UK, compliance ranged between 30-50% [26, 27]. Whether the relatively high TBE vaccine compliance observed here extends to other vaccines in Switzerland, though, is unclear.” (page 12, lines 213-231)

Interchangeability - main reference should be WHO Position paper, not a 2006 single reference. WHO Position Paper is not at all cited.

Thank you for this comment. To clarify, the 2006 reference is a review by Broker and Schondorf which is cited by the WHO position paper. While we cited the WHO position paper elsewhere (reference #1), we agree that it is correct to add it here as well. We also include a recent systematic review conducted by a collaborator which comes to the same conclusion (page 13, lines 240-242, lines 253-255). We hope this change adds weight and clarity to our statement.

---

## [Editor Report · Decision Letter 1]

3 Nov 2021

A Cross-Sectional Study Evaluating Tick-borne Encephalitis Vaccine Uptake and Timeliness Among Adults in Switzerland

PONE-D-21-05494R1

Dear Dr. Lang,

We’re pleased to inform you that your manuscript has been judged scientifically suitable for publication and will be formally accepted for publication once it meets all outstanding technical requirements.

Kind regards,

Ray Borrow, Ph.D., FRCPath

Academic Editor

PLOS ONE
---

## [Editor Report · Acceptance letter]

3 Dec 2021

PONE-D-21-05494R1 

A cross-sectional study evaluating Tick-borne Encephalitis vaccine uptake and timeliness among adults in Switzerland 

Dear Dr. Lang:

I'm pleased to inform you that your manuscript has been deemed suitable for publication in PLOS ONE. Congratulations! Your manuscript is now with our production department. 

Kind regards, 

on behalf of

Prof. Ray Borrow 

Academic Editor

PLOS ONE